# Optimization of a Newly Developed Chamber Setup for Spatial Dust Measurements in the Context of Containment

**DOI:** 10.3390/pharmaceutics17050565

**Published:** 2025-04-25

**Authors:** Hendrik Küllmar, Martin Schöler, Claudia S. Leopold

**Affiliations:** 1Division of Pharmaceutical Technology, University of Hamburg, Bundesstr. 45, 20146 Hamburg, Germany; hendrik.kuellmar@uni-hamburg.de; 2Fette Compacting GmbH, Grabauer Straße 24, 21493 Schwarzenbek, Germany; mschoeler@fette-compacting.com

**Keywords:** containment, dustiness chamber, IOM sampler, spatial dust distribution design of experiments

## Abstract

**Background/Objectives:** A specially designed chamber setup for containment investigations of pharmaceutical dusts was recently developed. The aim of the present study was to optimize the measurement procedure with this chamber setup, focusing on the atomization parameters. The optimization was aimed at a maximization of the amount of detected dust and a minimization of the required sample mass. **Methods:** For this purpose, the safe surrogate substance acetaminophen was used for dust measurements. In addition to the atomization parameters investigated by a design of experiments, the cleaning of the chamber setup and the selection of two different types of acetaminophen with different physicochemical properties were examined. **Results:** By altering the cleaning method of the chamber setup, more than a tenfold increase of detected acetaminophen was observed. In addition, by selecting the more appropriate acetaminophen type, the totally detected acetaminophen amount was further increased by more than 25%. By means of the design of experiments two models were developed, one dealing with the atomization parameters with regard to the atomization effectiveness and the other describing their influence on the spatial dust distribution of acetaminophen. Based on the model for atomization effectiveness, the totally detected acetaminophen amount may be increased by more than double at a constant sample mass. **Conclusions:** In summary, the measurement procedure of the chamber setup was optimized in terms of the cleaning method, surrogate choice, and the adjustment of the atomization parameters, giving valuable insights to deepen our understanding of dustiness and the spatial distribution of dust in the context of containment.

## 1. Introduction

Highly potent active pharmaceutical ingredients (HPAPIs) have lately become increasingly important in therapeutic fields such as oncology or the treatment of autoimmune diseases [1]. With high-throughput screening and the application of artificial intelligence, an increasing number of innovative and highly effective compounds have been developed [2,3,4,5]. Because of their high pharmacological and toxicological potential, HPAPIs often pose a risk for the personnel in the pharmaceutical industry during manufacturing [3,6]. To control the exposure of personnel to HPAPIs, so-called Occupational Exposure Limits (OELs) are implemented to ensure that this exposure does not exceed a tolerable level [7,8,9,10]. In addition, cross-contamination during manufacturing must be prevented, and environmental protection is of concern [6,11,12]. Because of the high risks posed by HPAPIs during the manufacturing of pharmaceuticals, containment measures have to be implemented at production sites, machines, and devices. These measures involve barrier concepts such as airlocks or housings for production machines as well as pressure differentials and high performance filter systems for ambient air [6,13,14,15].

The dustiness of a powder is defined as its propensity to generate airborne particles during processes such as mixing, milling, or transferring, which may lead to the generation of inhalable dust [14,16,17]. Airborne dust comprises particles with an aerodynamic diameter of up to 100 µm. The largest inhaled particles with an aerodynamic diameter of 100–30 µm are deposited between the point of entry (lips or nares) and the larynx. Smaller particles with a size of less than 10 µm may reach the thoracic region, and maximum deposition occurs with particles of approximately 2 µm aerodynamic diameter in the alveolar region. Particles smaller than 0.5 µm are usually exhaled without being deposited [17].

As dustiness with its associated health risks is of concern in different industries, various chambers for dustiness measurements of powders have been described in recent years [18,19,20,21,22,23,24,25,26]. Most of these systems require a relatively high amount of powder sample. As most active pharmaceutical ingredients (APIs) are of high production costs and thus of limited quantity, devices suitable for small sample quantities and/or pharmaceutical powders have been developed [27,28,29,30].

Dustiness is dependent on various physicochemical powder properties such as moisture content, particle morphology, density, and particle size distribution [28,31,32,33]. In addition, the type and amount of energy input needed for atomization as well as the flight path from the atomization to the detection site may influence the measured dustiness [18,32].

Because of the above-mentioned health risks associated with the handling of HPAPIs, the International Society for Pharmaceutical Engineering (ISPE) recommends safe surrogate substances such as lactose, naproxen sodium, or acetaminophen for tests of containment equipment and pharmaceutical dustiness investigations in the laboratory [10,13,34]. The choice of the surrogate substance should be based on its limits of analytical quantification as well as its physicochemical powder properties. For example, acetaminophen is a suitable surrogate for HPAPIs of low molecular weight. Furthermore, the quantification limit of acetaminophen by HPLC with subsequent spectrophotometrical detection is low, making it an industry-accepted standard for dustiness investigations [10,35,36]. If no suitable surrogate exists for dustiness investigations of a specific HPAPI, the risk posed for employees by handling this HPAPI has to be assessed. The risk assessment includes the identification of any potential hazards, defining who may be harmed, and the verification of whether existing precautions are adequate or need improvements [1]. These precautions may include, among others, the use of bottles with split butterfly valve connectors, handling of HPAPIs in isolators, as well as personal protective equipment [1,13].

In a previous study, a newly developed chamber setup was introduced, being the first of its kind, allowing not only the investigation of the dustiness of an API but also the determination of its spatial dust distribution. The results of these investigations were verified with computational fluid dynamics [37]. As the detected amount of generated airborne dust was low compared to the applied sample amount, the objective of this study was to optimize the presented chamber setup regarding its atomization effectiveness. This was performed by alteration of the cleaning method and by surrogate choice. As mentioned above, the type and amount of energy input applied for atomization of a powder sample is of major concern in dustiness investigations. To our knowledge, most investigations and comparisons of dustiness chambers comprised different methods for atomization of powder samples (for example, a rotating drum or a single-drop method). In addition, these studies only focused on experimental parameters such as sample mass, sampling duration, or volume flow rate for filter-based sampling, while none of these chambers were capable of measuring the spatial distribution of pharmaceutical dust [18,26,27,29,30,32]. Therefore, in the present study, the influence of the atomization parameters for pneumatic atomization with a newly developed chamber setup on the detected dustiness was investigated by application of a design of experiments approach.

## 2. Materials and Methods

### 2.1. Materials

The determination of the spatial distribution of dust was carried out with two different types of acetaminophen as safe surrogate substances (ACAM 1: Caelo, Hilden, Germany; ACAM 2: Fagron, Glinde, Germany), differing in their particle size and shape.

### 2.2. Methods

#### 2.2.1. Chamber Setup

The experimental setup was described in detail in a previous publication [37]. Briefly, the chamber setup consisted of two connected chambers, a small emission chamber and a larger detection chamber made of acrylic glass (Plexiglas XT, Polyvantis, Weiterstadt, Germany), as depicted in Figure 1. The emission chamber showed a volume of approximately 7.3 L and was connected to the detection chamber at one of nine possible positions by a protruding tube fixed to the emission chamber. The detection chamber with a volume of 1 m^3^ was set up in two different arrangements, either with oppositely or with orthogonally positioned measurement spots. In the present study all experiments were carried out with the orthogonal arrangement as displayed in Figure 1.

The measurement spots for dust detection were equipped with nine IOM samplers (I.O.M. Multi-Dust samplers, SKC, Blandford Forum, UK), each connected to an air sampling pump (Flite 4, SKC, Blandford Forum, UK) with independent flowmeters being located in the tube connections, as shown in Figure 2. A double-acting ball valve was connected to an orifice with an attachment point located in the top plate of the emission chamber and used for atomization of the powder samples. The ball valve was pneumatically controlled by a 5/2-way solenoid valve. An arrangement of two 3/2-way solenoid valves and two 3/2-way pneumatic valves controlled the duration of the airflow through the IOM samplers and the flow of pressurized air used for atomization (Figure 2). For the present study, three modifications with regard to the initial chamber setup were made. A compressor (Metabo Mega 400-50W, Nürtingen, Germany) was used as a source for compressed air for atomization to generate pre-pressures of up to 10 bar. To regulate the high pressures, the pressure-reducing valve (Futura 1 RP14-10 F, Riegler, Bad Urach, Germany) had to be exchanged. To make use of these pressures for high atomization volume flow rates, the flowmeter (FR4500, Key Instruments, Croydon, PA, USA) installed upstream of the double-acting ball valve also had to be exchanged. A Programmable Logic Controller (PLC: Siemens LOGO! 12/24 RCE, Munich, Germany) controlled all valves of the chamber setup. To dissipate electrostatic charges, a grounding cable was attached to the chamber setup.

#### 2.2.2. Dust Measurements

For each dust measurement, the respective sample was transferred into a small tube above the double-acting ball valve. After starting the measurement by actuating a toggle switch on the PLC, the programmed sequence led to the immediate opening of the double-acting ball valve (Figure 2). Thereby, the sample was atomized with compressed air, introduced into the emission chamber, and passed through the connecting orifice from the emission to the detection chamber. As the lag time between atomization and detection was set to 0 s, the 3/2-way solenoid valve between the IOM samplers and the vacuum pump was activated simultaneously. By this measure, an immediate airflow through each of the glass microfiber filters (1820-025, Whatman, Little Chalfont, UK) fixed within the IOM samplers with a flow rate of 2 L/min was realized for 278 min. Thereby, the ACAM dust moved via convective and diffusive mass transport and eventually settled on the filters. This settled amount was quantified by HPLC.

#### 2.2.3. HPLC Assay

The quantification of the settled amount of ACAM on the glass microfiber filters was performed by HPLC (Chromaster 5000, VWR-Hitachi, Tokyo, Japan) using a LiChroCART RP-18e 250-4 column (Merck, Darmstadt, Germany). Therefore, the filters were withdrawn from the IOM samplers and transferred to individual iodine flasks. The cassette fronts of each of the IOM samplers were swabbed with additional glass microfiber filters. These filters were also transferred to the corresponding iodine flasks. ACAM was extracted from the filters with 2 mL of the mobile phase, consisting of a mixture of acetonitrile and water (75:25 *v*/*v*), and adjusted to a pH of 3.5 with phosphoric acid. Therefore, the flasks were shaken for 30 min at 100 rpm (Unimax 1010, Heidolph Instruments, Kelheim, Germany). The obtained extracts were diluted with the mobile phase (1:150), and approximately 1.5 mL of the dilutions were transferred to HPLC vials. 20 µL of each diluted sample was injected into the sample loop, and the HPLC flow rate was set to 1 mL/min. The measurements were carried out at a temperature of 22 °C, and the spectrophotometrical detection was performed with a Diode Array Detector (Chromaster 5430, Hitachi, Tokyo, Japan). Because of the higher ACAM load on the filters, caused by the optimization of the dust measurements performed in the present study, the calibration range had to be extended compared to a previous study [37] and was linear between 0.033 µg/mL and 19.99 µg/mL (R^2^ = 0.9999).

#### 2.2.4. Cleaning of the Chamber Setup

Two different methods for cleaning the chamber setup were investigated. In “Cleaning method I”, the residual ACAM dust deposited on the inner panel surfaces was removed with a vacuum cleaner (Nilfisk GM 80, Bellenberg, Germany). Subsequently, the surfaces were wiped with a moistened microfiber cloth. The residual moisture was removed with Kimtech Science Precision Wipes (Kimberly-Clark, Koblenz, Germany). In “Cleaning method II”, the residual ACAM dust was directly removed with a moistened microfiber cloth. The residual moisture was removed with a conventional hair dryer (ThermoProtect 2200 W, Philips, Hamburg, Germany) to prevent electrostatic charging of the chamber setup as a result of friction. The electrostatic potential of the acrylic panel surfaces of the emission and detection chamber was measured with an IZH10 Handheld Electrostatic Meter (SMC Corporation, Tokyo, Japan) prior to cleaning and after Cleaning method I and II.

To evaluate the influence of the cleaning methods on the dust measurements, samples of ACAM 2 with a mass of 250 mg each were measured. The compressor was set to 10 bar, and the pre-pressure at the pressure reducing valve was reduced to 6 bar. The volume flow rate for atomization was set to 300 L/min with a duration of 0.2 s.

#### 2.2.5. Sieving of the Surrogate

Both ACAM types were sieved with 90 µm mesh size (LINKER Industrie-Technik, Kassel, Germany), as this step may improve the reproducibility of dust measurements [38]. By this means a coarse and a fine fraction were obtained for each ACAM type. The coarse fractions were discarded, and only the fine fractions were used in this study.

#### 2.2.6. Particle Size Distribution

The particle size distributions of both ACAM types were determined in triplicate by laser diffractometry (Helos KR, Sympatec, Clausthal-Zellerfeld, Germany). The measurements were performed with a lens with an effective range of 0.5–875 µm. Compressed air at a pressure of 1.5 bar was used to disperse the ACAM powder samples. The particle size distribution was analyzed with the Paqxos software (Version 2.0.3, Sympatec, Clausthal-Zellerfeld, Germany).

#### 2.2.7. Residual Moisture Content

The residual moisture contents of both ACAM types were measured in triplicate via thermal gravimetric analysis (Pyris 1 TGA, PerkinElmer, Waltham, MA, USA). Therefore, approximately 15 mg of ACAM were heated up with a heat rate of 10 °C/min to 105 °C and were kept at this temperature for 30 min to remove the residual moisture.

#### 2.2.8. True Density

The true densities of the two ACAM types were measured with a helium pycnometer (Pycnomatic ATC EVO, Porotec, Hofheim am Taunus, Germany). For this purpose, approximately 4 g of each ACAM type, measured in triplicate, were used, and five measurement cycles per sample were performed.

#### 2.2.9. Bulk Density

In accordance with the monograph 2.9.34, “Bulk Density of Powders” in the European Pharmacopoeia [39], the untapped and tapped bulk densities of both ACAM types were determined with a jolting volumeter (STAV 2003, J. Engelsmann, Ludwigshafen, Germany) in triplicate. Therefore, approximately 22 g of ACAM 1 and approximately 40 g of ACAM 2 were transferred into a 100 mL graduated measuring cylinder and subjected to the specified number of taps. The determined volumes were used to calculate the Hausner ratios.

#### 2.2.10. Scanning Electron Microscopy (SEM)

To investigate the particle shape of the two ACAM types, SEM images of the particles were taken with a Scanning Electron Microscope (Zeiss EVO MA10, Carl Zeiss, Oberkochen, Germany). The measurements were carried out in a high vacuum with an acceleration voltage of 5 kV.

#### 2.2.11. Atomization Parameters (DoE)

The experimental design for the statistical and factorial analysis of the atomization parameters was created with the software Design-Expert (Version 11.1.2.0, Stat-Ease Inc., Minneapolis, MN, USA). An I-optimal design was chosen to analyze the effect of three atomization parameters (atomization duration, atomization volume flow rate, and ACAM sample mass). A custom optimal design had to be chosen because the volume flow rate of the flowmeter used for atomization could only be set in increments of 10 L/min. From the given options, the I-optimal design was selected, as designs based on the I-optimality criterion minimize the integral of the prediction variance across the design space [40,41]. This design is commonly used if the response surface has to be modeled with good precision and if the significance of the investigated factors has been determined [40,42,43]. For analysis of the three atomization parameters mentioned above, for the basic model, ten model points were required. Five additional lack-of-fit points were added if a higher-order model, such as a quadratic model, was required to describe the response surface. Eight replicate points were added to estimate the precision of the model. Another eight additional model points were added to improve the statistical power (>80%) of potential two-factor interactions.

The choice of the factor limits for the DoE design is based on preliminary tests and a previous study and is shown in Table 1 [37]. Based on this previous study, the upper limit for the atomization duration was set to 5 s, as with this duration it was possible to obtain sufficient airborne dust concentrations in the detection chamber of the setup. The lower duration was set to 0.1 s because a synchronization of the double-acting ball valve and the 3/2-way pneumatic valve was not possible with shorter atomization durations due to switching delays. The upper limit for the atomization volume flow rate was set to 300 L/min, representing the maximum adjustable flow rate for the flowmeter. The lower limit was set to 100 L/min, as already applied in the above-mentioned previous study. ACAM 1 (50 to 500 mg) was selected for the DoE because of its higher dustiness (see Section 3.3). The compressor was set to its maximum pressure of 10 bar, which was subsequently reduced to 4 bar by the pressure-reducing valve. In a preliminary study, the reduction to 4 bar turned out to be best suited to achieve a constant volume flow over several seconds for both high and low atomization volume flow rates with the flowmeter. With a pressure reduction to values lower than 4 bar, the pressurized air was insufficient to achieve a volume flow rate of 300 L/min, and with a reduction to values higher than 4 bar, the compressed air of the compressor was not sufficient to maintain a constant volume flow for a duration of 5 s.

A model for the atomization effectiveness of the chamber setup was created by selecting the totally settled amount of ACAM 1 on all nine filters within the IOM samplers as a response. A further model for the spatial dust distribution in the detection chamber was created by selecting the mathematical difference of the settled amount of ACAM 1 between the lower row of IOM samplers (#7–#9) and the upper row of IOM samplers (#1–#3) as response, named ‘Mass difference between lower and upper IOM samplers’ throughout this paper.

Both models were statistically evaluated based on an analysis of variance (ANOVA). With the ANOVA, the significance of the models, their lack-of-fit, their R^2^, adjusted R^2^, and predicted R^2^ were determined. Non-significant model terms were omitted unless required for support of the factor hierarchy.

For the validation of both models, two additional measurements (confirmation runs) each were performed in triplicate at factor combinations different than those used to calculate the models. One confirmation run was carried out at the center point of the design space with an atomization duration of 2.55 s, an atomization volume flow rate of 200 L/min, and an ACAM 1 sample mass of 275 mg. For the other confirmation run, the atomization parameters were randomly chosen in the design space with an atomization duration of 3 s, an atomization volume flow rate of 100 L/min, and an ACAM 1 sample mass of 500 mg.

## 3. Results and Discussion

### 3.1. Powder Characterization

Various factors may influence the dustiness of powders. The particle size distribution and the residual moisture content play a major role; however, other parameters such as the bulk density and particle shape may also be of importance [28,32,33]. These relevant powder properties were determined for both investigated ACAM types used as surrogates in this study. The respective values are shown in Table 2. Even though both ACAM types were sieved with 90 µm mesh size, they differed in their particle size distribution. ACAM 2 showed larger particle sizes than ACAM 1.

For both ACAM types, no residual moisture content was detectable, and therefore, its supposedly high influence on dustiness was ruled out. The same applied to the true density, which did not show a significant difference (*p* < 0.05) regarding both ACAM types. Hausner ratios of 1.78 for ACAM 1 and 1.66 for ACAM 2 indicated a very, very poor powder flowability of both ACAM types [44]. ACAM 2 showed higher untapped and tapped bulk densities than ACAM 1.

In Figure 3, SEM images of ACAM 1 (a) and ACAM 2 (b) are shown. It is clearly visible that the particles of the two different types of ACAM exhibited different shapes.

The particles of ACAM 1 showed a more granular, almost spherical shape, whereas the particles of ACAM 2 exhibited a more elongated, irregular, needle-like shape.

### 3.2. Influence of the Cleaning Method

The totally settled amount of ACAM 2 after each of the two cleaning methods was measured, and the results are shown in Figure 4. A highly significant difference (*p* < 0.0001) was evident, with a more than tenfold higher totally settled amount of ACAM 2 if the chamber setup was cleaned with Cleaning method II as compared to Cleaning method I. This result demonstrated a strong influence of the cleaning method on the dustiness measurements. By drying the acrylic glass panels of the chamber setup with Kimtech Science Precision Wipes as described in Cleaning method I, electrostatic charging of the panels, especially with the attached grounding cable, was not expected. However, dust traces remained on the panels in the form of wipe marks after each experiment. ACAM itself shows a high tendency for electrostatic interactions with processing vessels, and acrylic glass exhibits a low conductivity [45,46,47]. Therefore, the presence of wipe marks combined with the results of the cleaning experiments strongly suggests electrostatic charging of the acrylic panels, resulting in a loss of airborne dust. Generally, electrostatic charging is described as one major factor influencing powder dustiness [31]. The lower dustiness detected after application of Cleaning method I was therefore assumed to be caused by electrostatic charging. Thus, the electrostatic potential of the emission and detection chamber was determined prior to cleaning, after Cleaning method I, and after Cleaning method II. The electrostatic potential of the emission chamber surface prior to cleaning amounted to 0.16 ± 0.12 kV, after Cleaning method I to 7.34 ± 0.49 kV, and after Cleaning method II to −2.40 ± 0.21 kV (*n* = 5). Similar values were measured at the surface of the detection chamber. The electrostatic potential prior to cleaning resulted in −0.01 ± 0.01 kV, after Cleaning method I in −1.56 ± 0.32 kV, and after Cleaning method II in 7.52 ± 0.61 kV (*n* = 5). The results demonstrated a significant difference (*p* < 0.0001) in the electrostatic potential between the two cleaning methods, verifying their influence on the detected settled amount of ACAM in this study.

With the change to a friction-free cleaning method, it was possible to mitigate electrostatic charges on the panels. To further reduce electrostatic charges, Sayahi et al. described an anti-static system consisting of a helical pattern of grounded copper strips and wire attached to their chamber setup, which was also made of acrylic glass [48]. With the modified cleaning method, a potential bias of the chamber setup when comparing electrostatically chargeable powders with non-chargeable substances is reduced, thereby improving its reliability. The results underline the importance of mitigating electrostatic charges when investigating dustiness in a laboratory environment. However, electrostatic charging only plays a minor role in pharmaceutical manufacturing under contained conditions, as production machines are usually grounded. Furthermore, tablet presses or isolators for the manufacture of dosage forms containing HPAPIs may be washed in place.

### 3.3. Comparison of the ACAM Types

The difference in the totally settled amount of ACAM 1 and ACAM 2, obtained with the same atomization parameters as used for the cleaning evaluation, is also shown in Figure 4. As the Cleaning method II led to higher dustiness values, as mentioned above, it was selected for the comparison of the two ACAM types. A significant difference (*p* < 0.05) between the totally settled amount of ACAM 1 and ACAM 2 was evident. As mentioned in the introduction, if investigating the dustiness of powders, the energy input needed for atomization must be taken into consideration [18]. As the atomization parameters for the comparison of both ACAM types were identical and no residual moisture content as well as no significant difference in their true density was detectable, the observed dustiness results were caused most likely by their different particle size distributions, particle shapes, and bulk densities.

Several authors identified the particle size distribution as one of the most important powder characteristics affecting dustiness [28,32,36,38,49,50]. Regarding the particle size, Plinke et al. postulated that the size-specific dust generation rate increases with the particle diameter *d*. They explained this observation in their experiments with the fact that the separation forces, such as impaction, increase with *d*^3^, whereas binding forces based on van der Waals attraction and capillary forces are proportional to *d* [32]. This influence of binding and separation forces is overlaid by the particle transport away from the atomization site. Larger particles (>10 µm) are affected by gravity and thus sedimentation, preventing the transport from the atomization site to the measuring site [32]. Results of coating experiments with powder particles on an aluminum tray indicated that with decreasing particle size, dustiness tended to increase [49]. This observation is supported by findings of Wirth et al. in two studies, where different particle size fractions of ACAM were investigated regarding their dustiness. A significant decrease in dust emission was observed with increasing particle size [36,51]. This is in accordance with the detected differences in dustiness observed with the two ACAM types in the present study.

Beyond that, the particle morphology may affect the measured dustiness because of its influence on interparticular adherence as well as attraction to or repulsion from surfaces [24,28,51]. As shown in Figure 3, ACAM 2 showed a more elongated, irregular, needle-like shape. In experiments with nanoparticles carried out by O’Shaughnessy et al., a lower dustiness for fibrous particles in comparison to granular particles was demonstrated [28]. This result supports the lower dustiness of ACAM 2 observed in this study. Furthermore, a significantly higher untapped bulk density (*p* < 0.01) of ACAM 2 than that of ACAM 1 was observed (Table 2). This also may have contributed to the lower dustiness of ACAM 2 [31].

### 3.4. Results of the DoE

An I-optimal design was used to analyze the effect of the three atomization parameters on the atomization effectiveness and spatial dust distribution within the detection chamber of the newly developed chamber setup. The results are shown in Table 3. The non-significant lack of fit showed that both models described the original data well. One value had to be excluded from the data set of the spatial dust distribution model because of a sampling error. This did not affect the totally settled ACAM amount and thus the model for the atomization effectiveness. For this model, the high values for R^2^, adjusted R^2^, and predicted R^2^, as well as the high adequate precision value, indicated a good model for correlation and prediction. These values were lower for the spatial dust distribution model, but the predicted R^2^ was still in reasonable agreement with the adjusted R^2^ (difference <0.2), and the adequate precision was higher than 4. Therefore, the model was used to navigate the design space. As shown in Table 3, non-significant model terms and higher-order models were excluded, if possible, to simplify the models.

The results for the confirmation runs are listed in Table 4. The predicted values of the statistical models were compared to the measured values of the confirmation runs. For the atomization effectiveness model, a good overall prediction was achieved with deviations of only 3.49 and 16.67%, respectively. However, the second model for the spatial dust distribution failed in terms of the confirmation runs at the center point of the model with a deviation of 120.59% from the predicted value. This was caused by slightly more settled dust on the filters in the upper IOM row than in the lower IOM row. The second combination of atomization parameters resulted in a deviation of 47.12%, indicating an imprecise overall model prediction. Therefore, predictions based on this model should only be made with keeping the quality of the model in mind. On the one hand, the imprecise prediction was caused by the chosen factor limits. These limits were selected based on preliminary studies with the optimization of the atomization effectiveness in mind. On the other hand, as the main goal of this study was the optimization of the atomization effectiveness, ACAM was sieved to improve the reproducibility of the dustiness measurements [38]. This sieving step narrowed the particle size distribution (Table 2) by removing the coarse fraction of the particles. Consequently, the fraction of particles affected by sedimentation within the detection chamber was reduced, and thus the spatial dust distribution with regard to the mass difference between lower and upper IOM samplers was less pronounced.

In Figure 5 the results of the response surface model for the atomization effectiveness are shown. As the sample mass did not show any factor interactions (Table 3), its direct influence on the totally settled ACAM amount is shown in Figure 5a. It is obvious that the curve profile of the settled ACAM amount shows a linear tendency from 50 mg up to approximately 300 mg with a peak maximum of 410 mg. A similar effect was observed with a different chamber setup for dustiness investigations using an identical double-acting ball valve for atomization [30]. This observation indicates that with this ball valve geometry, the atomization of even low sample amounts is possible, which is advantageous with regard to reduced experimental costs. Furthermore, by adjusting the atomization duration and volume flow rate according to the results of the DoE, the totally settled ACAM amount may be more than doubled at a constant sample mass (Figure 5b). This poses a major advantage compared to a previous study [37].

The response surface shown in Figure 5b describes the two-factor interaction of the atomization volume flow rate and the atomization duration. It is evident that the totally settled ACAM amount clearly increases if an atomization duration of over 2 s is combined with an atomization volume flow rate of more than 150 L/min. By combining these results, a high, totally settled ACAM amount may be achieved with sample masses as low as possible.

The results of the response surface model for the mass difference between lower and upper IOM samplers are shown in Figure 6. In Figure 6a, a linear relationship between the sample mass and the corresponding mass difference becomes apparent, similar to the linear tendency of the totally settled ACAM amount up to a sample mass of 410 mg, as shown in Figure 5a. This observation is especially of interest, as both the atomization effectiveness and the spatial dust distribution may be increased by selecting higher sample amounts. With the ANOVA of the spatial dust distribution model, the same two-factor interaction of atomization volume flow rate and atomization duration as with the ANOVA of the atomization effectiveness model was recognized (Table 3). The two-factor interaction shown in Figure 6b indicates that the spatial dust distribution may be increased by increasing both the atomization duration and the atomization volume flow rate. Thus, both models highlight the strong influence of the type and amount of applied energy for atomization—in this case compressed air—on dustiness and spatial dust distribution.

## 4. Conclusions

A newly developed chamber setup has been optimized with regard to the atomization parameters and other experimental factors. The detected amount of the surrogate acetaminophen was increased more than tenfold by an alteration of the cleaning method comprising a friction-free thermal drying. In addition, the choice of an appropriate acetaminophen type with regard to the particle size distribution and particle morphology resulted in a further increase of the detected surrogate amount. With an I-optimal design of experiments, a response surface model for the atomization parameters (atomization duration, atomization volume flow rate, and sample mass) and the resulting atomization effectiveness was created. According to this model, the atomization effectiveness may be improved by longer atomization durations, higher volume flow rates, and sample masses up to 400 mg. These results allow a significant decrease in the required sample mass and thus cost for dust investigations by increasing the atomization duration and volume flow rate. A second model with regard to the spatial dust distribution was designed based on the same dataset gathered for the optimization of the atomization effectiveness. However, it became apparent that the parameter limits for the design space selected to maximize the atomization effectiveness were poorly suited for the investigation of spatial dust distributions. This selection of factor limits and the use of sieved acetaminophen powder resulted in an imprecise response surface model. As the measurement procedure and the atomization parameters were optimized with regard to the detected surrogate amount in the present study, future studies will deal with the optimization of the determination of spatial dust distributions. For this purpose, non-sieved acetaminophen as well as more appropriate factor limits for the design space will be selected, and the results will be analyzed with a machine learning approach to better capture the complexity of spatial dust distribution. The chamber setup will then be used to investigate and compare the dustiness and spatial dust distribution of APIs with different physicochemical properties, such as particle morphology or particle size distribution. Beyond that, binary mixtures of an API and different excipients will be investigated. These studies may provide valuable insights into dustiness and especially the spatial distribution of pharmaceutical dusts, and based on this knowledge, the classification of HPAPIs to OELs is going to be more accurate and will further improve safety in the manufacturing of pharmaceutical dosage forms.

## Figures and Tables

**Figure 1 pharmaceutics-17-00565-f001:**
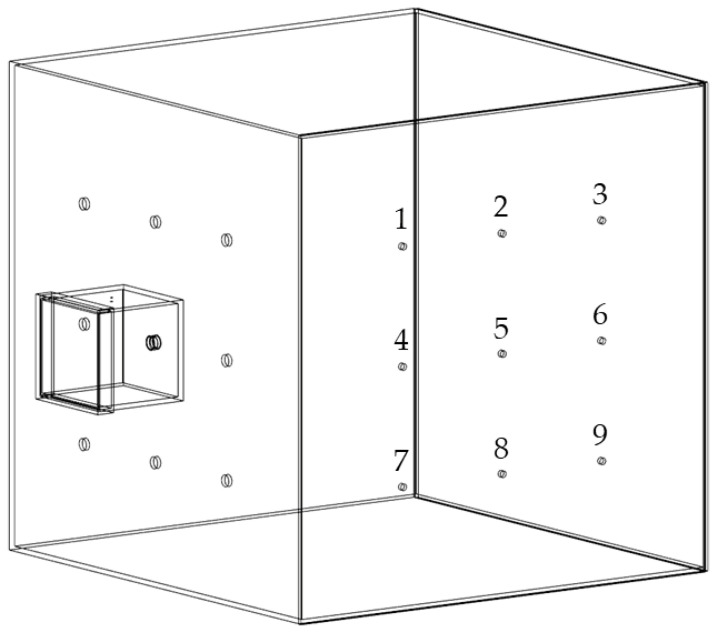
Simplified illustration of the chamber setup for dust measurements in orthogonal arrangement with measurement spots 1–9, modified after [37].

**Figure 2 pharmaceutics-17-00565-f002:**
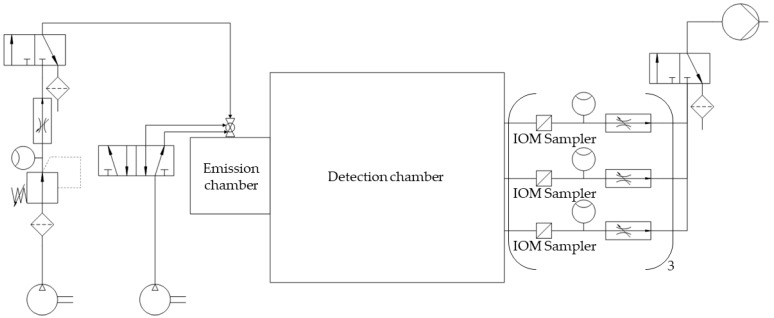
Simplified piping and instrumentation diagram of the optimized chamber setup.

**Figure 3 pharmaceutics-17-00565-f003:**
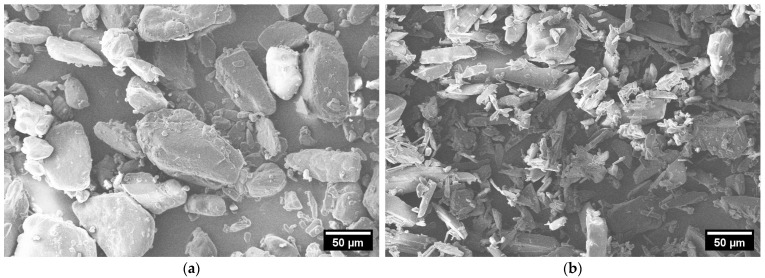
SEM images of ACAM 1 (**a**) and ACAM 2 (**b**).

**Figure 4 pharmaceutics-17-00565-f004:**
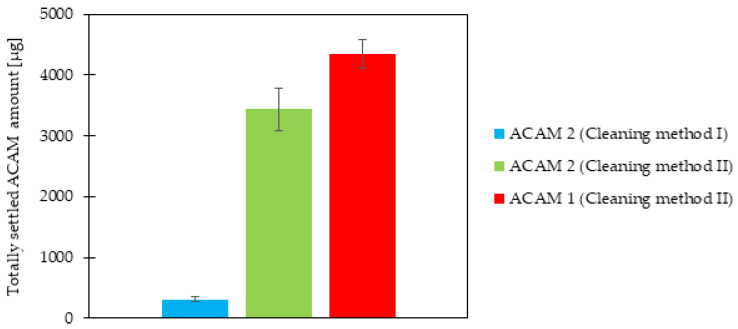
Totally settled amount of ACAM 2 after Cleaning methods I and II as well as that of ACAM 1 after Cleaning method II, on all nine filters within the IOM samplers (means ± SD, *n* = 3).

**Figure 5 pharmaceutics-17-00565-f005:**
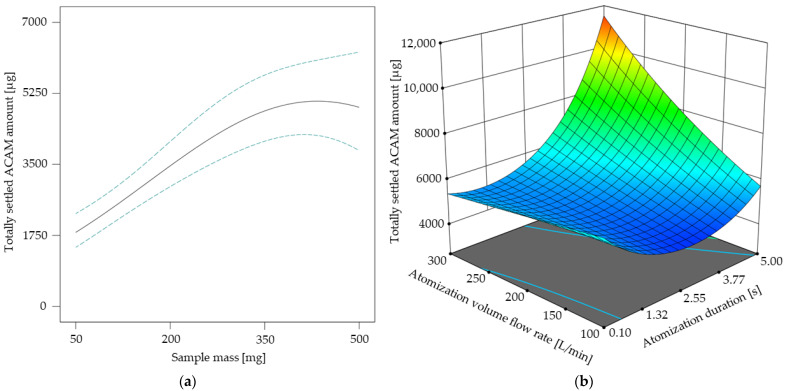
Totally settled amount of acetaminophen [µg] in dependence of the sample mass with its 95% CI bands at an atomization duration of 2.55 s and an atomization volume flow rate of 200 L/min (**a**). Response surface for the totally settled ACAM amount [µg] in dependence of the two-factor interaction at a sample mass of 410 mg (**b**).

**Figure 6 pharmaceutics-17-00565-f006:**
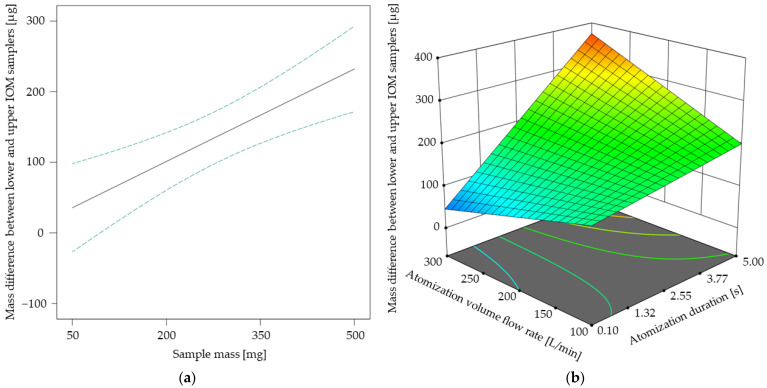
Mass difference between lower and upper IOM samplers [µg] in dependence of the sample mass with its 95% CI bands at an atomization duration of 2.55 s and an atomization volume flow rate of 200 L/min (**a**). Response surface for the mass difference between lower and upper IOM samplers [µg] in dependence of the two-factor interaction at a sample mass of 410 mg (**b**).

**Table 1 pharmaceutics-17-00565-t001:** Selected factor limits for the DoE.

Factor	Lower Limit	Upper Limit
Atomization duration [s]	0.1	5
Atomization volume flow rate [L/min]	100	300
ACAM 1 sample mass [mg]	50	500

**Table 2 pharmaceutics-17-00565-t002:** Physicochemical properties of the investigated surrogate ACAM (means ± SD, *n* = 3).

Powder Properties	ACAM 1	ACAM 2
Particle size [µm]		
x_10_	3.01 ± 0.08	4.54 ± 0.10
x_50_	16.84 ± 0.11	31.52 ± 0.18
x_90_	58.43 ± 0.59	76.69 ± 0.24
True density [g/cm^3^]	1.2903 ± 0.0043	1.2894 ± 0.0025
Untapped bulk density [g/mL]	0.233 ± 0.006	0.429 ± 0.002
Tapped bulk density [g/mL]	0.415 ± 0.011	0.710 ± 0.006
Hausner ratio	1.78 ± 0.01	1.66 ± 0.02

**Table 3 pharmaceutics-17-00565-t003:** ANOVA results of the DoE models.

Model	Atomization Effectiveness (Totally Settled Amount) *p*-Value	Spatial Dust Distribution (Lower–Upper IOM Row) *p*-Value
Model	0.0001	0.0001
Sample mass (A)	0.0001	0.0004
Atomization duration (B)	0.0204	0.0009
Atomization volume flow rate (C)	0.0357	Nonsignificant
Interaction AB	Nonsignificant	Nonsignificant
Interaction AC	Nonsignificant	Nonsignificant
Interaction BC	0.0103	0.0312
A^2^	0.0027	Nonsignificant
B^2^	0.0079	Nonsignificant
C^2^	Nonsignificant	Nonsignificant
Lack of fit	Nonsignificant	Nonsignificant
R^2^	0.8455	0.6525
Adjusted R^2^	0.8069	0.5969
Predicted R^2^	0.7381	0.4268
Adequate precision	15.7408	13.1658

**Table 4 pharmaceutics-17-00565-t004:** Confirmation of the DoE models.

Model	Atomization Effectiveness		Spatial Dust Distribution	
Atomization duration [s]	2.55	3	2.55	3
Atomization volume flow rate [L/min]	200	100	200	100
Sample mass [mg]	275	500	275	500
Predicted mean [µg]	4245.83	4294.13	133.95	220.26
Measured mean ± SD [µg] (*n* = 3)	4394.02 ± 184.32	5010.07 ± 359.31	−27.58 ± 2.54	116.48 ± 11.63
Deviation [%]	3.49	16.67	120.59	47.12

## Data Availability

Data is contained within the article.

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
