# Peer review of "Optimization of a Newly Developed Chamber Setup for Spatial Dust Measurements in the Context of Containment"

_pharmaceutics, 2025, doi:10.3390/pharmaceutics17050565_

Round 1

Reviewer 1 Report

Comments and Suggestions for Authors

This manuscript worked on optimizing chamber setup for measuring pharmaceutical dust containment. The author focusses on enhancing atomization effectiveness by adjusting key parameters. The study results demonstrate cleaning method and optimizing atomization settings significantly increased dust detection, improving the chamber’s effectiveness for containment investigations.

The manuscript presents a well-structured study with experimental design and data analysis. However, there are some areas that require clarification and additional explanation as listed in the comments below. I would accept this manuscript with minor revision.

  1. Minor revision on the grammar and sentence structure are recommended for clarity and readability.
  2. Author used specific atomization durations and air pressures but didn’t explain why these limits were selected. Please provide short explanation.
  3. Author need to provide strategies for mitigating electrostatic effects since it is making significant impact on the dust retention.
  4. It will be better to provide past studies and compare with present studies and show how optimized chamber setup improved from previously reported design for reliability, cost and efficiency.
Comments on the Quality of English Language

Need revision for more clarity and readability 

Reviewer 2 Report

Comments and Suggestions for Authors

Presented paper is devoted to optimize the previously developed chamber setup for spatial dust measurements regarding its atomization effectiveness. This was performed by alteration of the cleaning method and by application of a design of experiments approach investigating the atomization parameters.

1) The novelty of the work must be clearly addressed and discussed, compare your research with existing research findings and highlight novelty

2) Add quantitative research outcomes in the abstract section,

3) Conclusion section is missing some perspective related to the future research work

Reviewer 3 Report

Comments and Suggestions for Authors

The paper presents a study on the optimization of a newly developed chamber setup for spatial dust measurements in the context of containment, focusing on pharmaceutical dusts. While the study is well-structured and provides valuable insights, there are several points should be addressed before publication:

1. The study uses acetaminophen as a surrogate substance for dust measurements. While the authors mention that acetaminophen is recommended by the International Society for Pharmaceutical Engineering (ISPE), they do not provide a detailed justification for why acetaminophen was chosen over other potential surrogates (e.g., lactose, naproxen sodium). A more thorough discussion on the selection criteria, including the physicochemical properties of acetaminophen that make it suitable for this study, would strengthen the paper (physicochemical properties and relevance to HPAPIs).

2. Cleaning Method II showing a tenfold increase in detected acetaminophen compared to Cleaning Method I. why this difference occurs is limited. please provide experimental evidence with additional experiments.

3. model shows poor predictive performance; it should include a broader range of factor limits or using non-sieved acetaminophen to better capture the spatial distribution of dust.

4. add physicochemical properties that may influence dustiness.

5. the results may not fully translate to real-world pharmaceutical manufacturing environments. add a discussion on the practical implications of the findings for pharmaceutical manufacturing.

6. what is differences between acetaminophen and HPAPIs?

7. Consider using more advanced statistical techniques or machine learning models to better capture the complexity of dust distribution.

8. discuss the safety measures taken to protect researchers or the ethical considerations related to handling potentially hazardous materials, particularly when handling and measuring pharmaceutical dust.

9. Add more references from mdpi in recent years.

10. Add scale bars to the SEM images to indicate the magnification level for figure 3.

Round 2

Reviewer 3 Report

Comments and Suggestions for Authors

Pape can accept in current form.